# Extended Abstract Track

# Unsupervised learning of geometrical features from images by explicit group actions enforcement

**Luca Bottero**                                      luca.bottero192@edu.unito.it
and
**Valerio Pagliarino**                            valerio.pagliarino@edu.unito.it
*Via Pietro Giuria 1, 10126 Turin, Italy*

**Francesco Calisto**                          Francesco.Calisto@campus.lmu.de
*Theresienstr. 37, 80333 München, Germany*

**Editors:** Sophia Sanborn, Christian Shewmake, Simone Azeglio, Arianna Di Bernardo, Nina Miolane

## Abstract

In this work we propose an autoencoder architecture capable of automatically learning meaningful geometric features of objects in images, achieving a disentangled representation of 2D objects. It is made of a standard dense autoencoder that captures the *deep features* identifying the shapes and an additional encoder that extracts geometric latent variables regressed in an unsupervised manner. These are then used to apply a transformation on the output of the *deep features* decoder. The promising results show that this approach performs better than a non-constrained model having more degrees of freedom.

**Keywords:** Autoencoders, group actions, geometric priors, latent space disentanglement [1]

## 1. Introduction and motivation

Deep Neural Networks have widely proven to be an ideal tool to classify large datasets in various domains (see Alzubaidi et al. (2021) for a review), computer vision being one of them (Deng et al. (2009)). However, despite their numerous successes, sometime they lack generalisation and automatic meaningful feature extraction capabilities, which are effectively coherent with the task at hand. In addition, the leading paradigm for the training of NNs is supervised learning, which necessitates of costly labeling work. Therefore it is necessary to train such networks using datasets as large and varied as possible, but finding and constructing them with all necessary attention is usually expensive and their employment would possibly lead to unmanageable training costs.

Therefore, a recent direction of development in deep learning research is the investigation of new architectures possessing invariance properties with respect to specific transformations, endowing them with better generalization capabilities and a faster training process (see Cohen and Welling (2014) for a mathematical treatment of visual representation properties). A relevant example are architectures invariant with respect to some geometric transformations, encoded by a group action, which are able to process images by identifying objects independently of their location in space, similarly to what humans are able to do. If brought to a production stage, this approach could outperform standard data augmentation techniques, increasing the training efficiency considerably, similarly to other

---

1. All authors contributed equally

# Extended Abstract Track

Geometric Deep Learning models (Cohen and Welling (2016), Bronstein et al. (2021)).

In this specific work, we aim at constructing a deep learning architecture with the ability to disentangle roto-translational and scaling properties of objects in 2D images from the *intrinsic shape* of the object in a fully automated way. Being a theoretical study focussing primarily on geometrical aspects, we start by considering images with a single object at a time. This doesn't constrain the generality of the approach too much, because it is possible to embed this architecture in a pipeline where other methods are used for isolating single objects, such as *BOOST tracker*, *MIL tracker* and *CSRT Tracker* from the *OpenCV* library, Bradski (2000).

In particular, we present an architecture composed of two main ingredients: a deterministic autoencoder, which learns the intrinsic *deep features* representing the shape of the object in the image, and an encoder, capturing its position, orientation and size. After training the network, we apply the learned roto-translation matrix (representing the group action) to the output of the first autoencoder to reconstruct the original image. The idea of explicitly imposing a transformation rule is similar to the work carried out in Jaderberg et al. (2015), however here we are applying it to an unsupervised task on a fully connected autoencoder, highlighting the separation between geometric and deep features more sharply. With this technique, we are forcing the regression of geometrical features of the image only by imposing how such features will transforms the image as a whole after the decoding, making their learning completely unsupervised.

## 2. Model and training

### 2.1. Architecture

To set the ground for the rest of the article, let's agree that by *intrinsic* or *deep features* $\mathcal{S}_i$ we mean the shape and topology of the objects inside the image, and by *extrinsic* or *geometric features* $\Theta_j$ the ones relative to their immersion in the 2D image space, which encode an affine Lie group structure, as we will see.

Therefore we can write $\mathcal{S}_i \in \mathcal{L}_{int} \cong \mathbb{R}^{32}$, $\mathcal{L}_{int} = \{\text{shape}_1, \text{shape}_2, ...\}$, $i \in B$ for the former and $\Theta_j = (\theta_j, s_j, x_j, y_j) \in \mathcal{L}_{ext} \cong \mathbb{R}^4$, $j \in B$ for the latter, where $B$ is a subset of the input dataset (training batch).

On the one hand, the *geometric feature*s are learned by an encoder with a 4-neuron dense output layer, corresponding to the object's rotation angle $\theta$, scaling $s$ and position coordinates $(x, y)$. We take the modulus of the raw neural output of $\theta$ in order to improve convergence. On the other hand, the *deep* ones are encoded in a 32-dimensional latent space and decoded to the original image by a dense deterministic autoencoder. The $\Theta_j$ parameters are used to operate an affine transformation on the decoded image, represented by the following matrix:

$$M = \begin{pmatrix} s\cos\theta & -s\sin\theta & t_x s\cos\theta - t_y s\sin\theta \\ s\sin\theta & s\cos\theta & t_x s\sin\theta + t_y s\cos\theta \\ 0 & 0 & 1 \end{pmatrix} \ ,$$

given by composition of rotations, translations and scaling matrices. These represent affine invertible transformations from $V = \mathbb{R}^{64 \times 64}$ to itself, i.e. they belong to the associated

affine group Aff($V$). The latter are of the form $\phi(a) = ga + v$, with $a, v \in V$, $g \in GL(V)$, which means Aff($V$) $\cong V^{+} \rtimes GL(V)$ (Zimmermann and Kraft (2013)). Therefore Aff($V$) is a Lie group. Being a composition of Lie group action matrices, $M$ belongs to a Lie group itself.

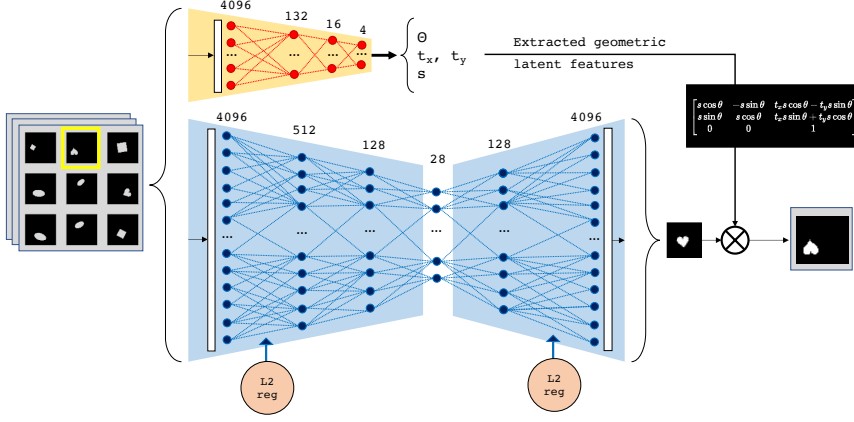

Figure 1: Schematic representation of the architecture

## 2.2. Training with a sample dataset

The network was trained on a batch of 10000 images sorted randomly from the dataset and the mean squared error (MSE) was used as a loss function: $l(x, y) = mean(\{(x_1 - y_1)^2, ..., (x_n - y_n)^2\})$. The minimization strategy is a first-order gradient-based optimization with adaptive learning rate (ADAM algorithm, Kingma and Ba (2014)), where the learning rate was initialised at $8e - 4$.

In order to test the disentangling capability of our model, we need a dataset with three properties: images that are simple enough to be reconstructed by a modest model with a decent accuracy in reasonable time; clearly different image shapes, image resolution high enough to ensure a relatively low quality degradation when the shapes are transformed.

Our choice is therefore the *dSprite* dataset (Matthey et al. (2017)), which is a good compromise satisfying these constraints. It is a synthetically created dataset built combining the three kinds transformations (namely rotation, translation and dilatation) applied in sequence on three shapes (hearth, ellipsis, square).

Our architecture has been implemented using the *PyTorch* framework and trained using 10 x Intel Xeon cores, 20 GB RAM and an NVidia Tesla T4 GPU with 16 GB of memory. The model was run for 50 iterations and reached a final loss of 4.19e-3 in 6 minutes and 28 seconds. The code can be found in the NeurReps-2022-public repository.

## 3. Results

A few examples of our results on validation set are shown in Fig.2. In the left canvas we compare the input image, the output of the dense autoencoder capturing the deep features and the final output image after group action application. The output seems to reproduce the original image accurately. The key point is that the autoencoder successfully learned the *deep* features of the objects separately from the extrinsic ones: this is proven by the fact that the AE output images are translated, rotated and scaled with respect to the output images. Moreover, AE latent spaces corresponding to different input images with the same shape are comparable, meaning shapes are detected correctly.

In the right part of Fig.2 we included some results obtained with a classic dense autoencoder with a larger number of degrees of freedom without any geometric enforcement: it is clear that the quality of the output image is considerably worse. It was trained in the same way and reached a loss of 7.58e-3 after 5 minutes and 25 seconds. This evidence supports the claim that our disentangled learning is profound and more accurate because a more efficient information compression in the latent space is performed.

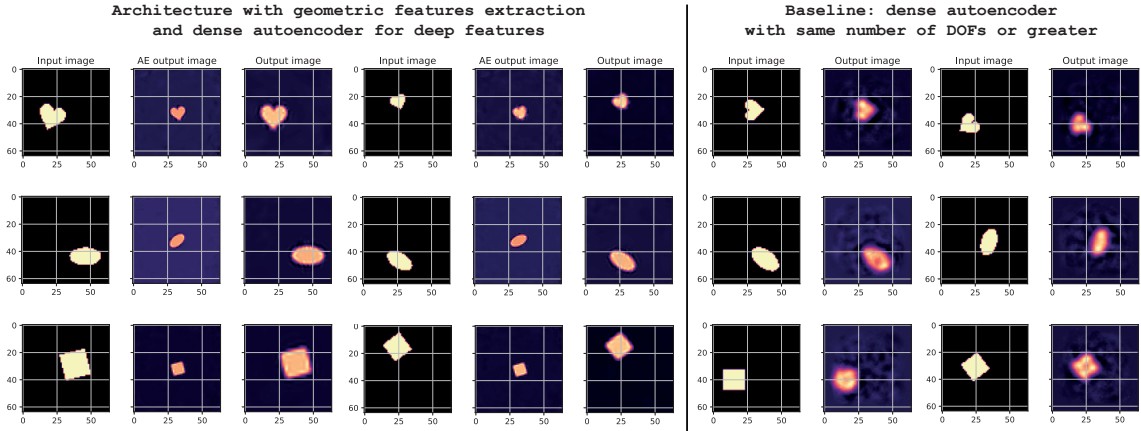

Figure 2: Results on validation dataset

## 4. Conclusions and outlook

We have devised a simple unsupervised architecture that was able to extract geometric features from images of 2D objects in an automated manner, by including a layer that enforces a group action. Additionally, other affine group structures could be enforced in the same manner, such as the physically-inspired Poincarè group, relevant to special relativity, providing meaningful insights on the problem at hand. We see the potential of our architecture to pave the way for an unsupervised partially-parametric generative model, which gives the user more control over the latent space and therefore on the generated output. In the context of Geometric Deep Learning, we see this work going in a direction parallel to Physics-Informed Machine Learning, where physical dynamics are imposed, instead of the geometry: in both cases, making the best out of the a-priori-known structures is a powerful

Extended Abstract Track

way to build more interpretable and efficient models, with generalization capabilities that elevate the process of learning to a more biologically-inspired endeavour.

## 5. Acknowledgement

First of all, we thank our colleagues Marco Nurisso and Luca Savant Aira for their contribution in the first conceptual developments of the work, and the Machine Learning Journal Club of the University of Turin for making this research work possible.
FC gratefully acknowledges the *Élite Graduate Course in Theoretical and Mathematical Physics* and the Élite Network of Bavaria for the support.
Furthermore, we are thankful to the University of Turin, HPC4AI and NPO Torino for their help.

## 6. Appendix

### 6.1. MSE loss vs iterations during the training and validation processes for the architecture with geometric features extraction

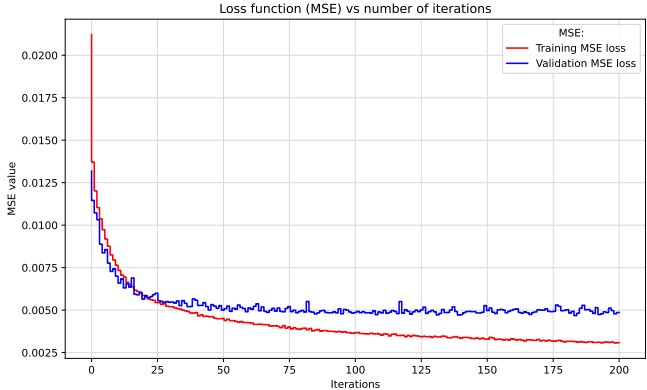

Figure 3: MSE loss vs iterations calculated with both training and validation datasets

### 6.2. MSE loss vs iterations during the training and validation processes for the unified autoencoder architecture

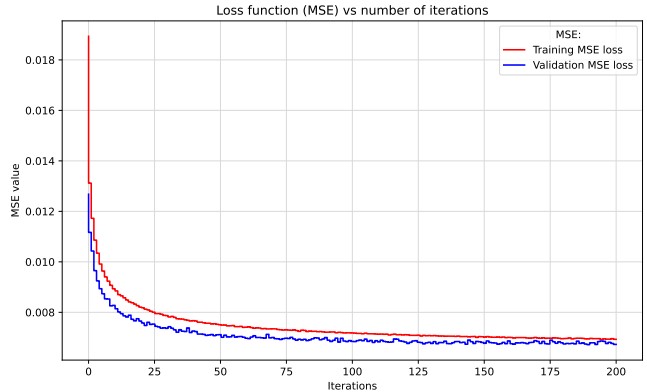

Figure 4: MSE loss vs iterations calculated with both training and validation datasets

Extended Abstract Track

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
