# OpenReview forum: "Unsupervised learning of geometrical features from images by explicit group actions enforcement"
_NeurIPS.cc/2022/Workshop/NeurReps — NeurReps 2022 Poster_

### Official Review · Reviewer_kg2u · 2022-10-06

**Confidence:** 3
**Soundness:** 2
**Presentation:** 1
**Contribution:** 2
**Overall Rating:** 3

**Summary:**

This work proposes an autoencoder with a structured latent space.  The latent space is separated into features that describe the geometry of the input shape and features that describe the shape itself.  Because the geometric latent features are used to parameterize an affine transformation, the latent space naturally disentangles itself using only a reconstruction loss on the output.  The idea is tested on the dSprite dataset where it qualitatively outperforms an autoencoder with an unstructured latent space.

**Questions:**

- Does the AE output image look the same when trained on different seeds? It's interesting that it predicts a small version of the shape in the middle of the image, but it seems coincidental.
- Is there a reason that an MSE loss was used instead of a binary cross entropy?  A cross-entropy loss may generate more crisp outputs since dSprites uses black-and-white images.
- Some of the network design decisions are not well justified.  Why are convolutional layers not used?  Why not use a variational auto-encoder?

**Limitations:**

It is unclear whether this approach would work for more complex images where the background changes or there are multiple objects present.  Moreover, for images with 3D objects, it would no longer be simple to parameterize a transformation of pixels to generate the output.

**Recommended Decision:**

2: Borderline

**Relevance:**

3: Solid fit

**Strengths And Weaknesses:**

Strengths
- The proposed network is able to learn a disentangled representation using only a reconstruction loss.

Weaknesses
- This work is missing a quantitative evaluation.  This could be fixed by reporting the mean squared error on the output.  Alternatively, dSprites provides labels for the geometric features learned by the proposed method.  Thus, it may be possible to evaluate how accurately the method learned these features.
- The idea of separately predicting the parameters for an affine transformation is not novel, and was proposed in Spatial Transformer Networks [1].
- The writing, especially in the introduction, could be improved to better convey the significance of this work.

[1] Jaderberg, Max, Karen Simonyan, and Andrew Zisserman. "Spatial transformer networks." Advances in neural information processing systems 28 (2015).


**Submission Track:**

Extended Abstract (4 Page)

---

### Official Review · Reviewer_8Sfz · 2022-10-11
**Very early steps towards unsupervised learning of geometric features from images applied to a trivial dataset.**

**Confidence:** 4
**Soundness:** 2
**Presentation:** 2
**Contribution:** 2
**Overall Rating:** 5

**Summary:**

This work proposes a way to indirectly learn 2D geometric features (i.e. angle, position, scale) of simple image inputs (dSprites; 3 classes) by minimizing the MSE between inputs and an affine transformation (using those geometric features) of a standard class template. These templates are the output of a deterministic auto-encoder that maps input instances of each class with any set of nuisance parameters (i.e. angles, positions, scales) to a fixed set. An MLP is used to learn these 2D geometric features and they are relative to the class template.

**Questions:**

- How does finding the 2D geometric features of input samples help useful downstream ML tasks?
- Would the approach work for a dataset where the generative factors go beyond shape plus affine transformations like rotation, scaling, and translation?

**Limitations:**

In the conclusion, the authors suggest that this line of work could be useful for physics-informed ML. However, the way I see it, is that ML is helping you determine the parameters of a hypothesis physical system. But the Physical system is not helping itself any Machine Learning downstream task.

**Recommended Decision:**

2: Borderline

**Relevance:**

3: Solid fit

**Strengths And Weaknesses:**

Although the work is motivated by 1) the problem of embedding invariances into networks to improve generalization and reduced reliance on data augmentations, and 2) the goal of learning disentangled representations, it does not really tackle any of these two problems.

The contribution of the work is simply to learn the 2D parameters of an affine transformation (known a-priori) of the inputs relative to a standard class template by minimizing a reconstruction loss. These parameters are –by definition– independent factors, and thus not follow the traditional referenced work on learning disentangled representations where latent axes are expected to represent independent factors of variation that are not known a-priori and one can sample from to generate novel samples.

Strengths:

- The authors learn 2D affine transformation parameters without specific supervision.

Weaknesses:

- The writing could be clearer. The flow of arguments leading to the proposed method was confusing. The explanation of the model and results could use a few more iterations to improve clarity. The title “unsupervised training with MSE loss” next to the inputs in Fig. 1 was confusing.
- There is little value in the proposed AE for “deep features” representing the shape as all it does is to recognize the input class and memorize a standard shape (an instance of the input with a fixed parameter set). This is because the target parameters the authors want to learn are those with which the dataset was generated in the first place (plus shape).

**Submission Track:**

Extended Abstract (4 Page)

---

### Official Review · Reviewer_LPFu · 2022-10-13
**Interesting preliminary results**

**Confidence:** 3
**Soundness:** 3
**Presentation:** 3
**Contribution:** 3
**Overall Rating:** 5

**Summary:**

The authors propose an autoencoder architecture capable of automatically learning meaningful geometric features of objects in images. In particular, they propose a scheme that disentangles the representation of 2D objects into features that identify the shapes and features that  extract geometric latent variables such as rotation angle, position and scaling. They apply their proposed scheme in a synthetic, shape based dataset, and they show that the obtained results are aligned with the intuition.



**Questions:**

1) It is not clear from the text how from the reconstruction of the autoencoder you get an image that is aligned (representative of that class- see Fig 1 and the image of the heart). That point should be clarified better as it is a core component of the algorithm.  Does it generalize to more complicated shapes?

2) If you have partial knowledge of the transformation matrix M, would the proposed method still work?



**Limitations:**

The main limitation of the work, that is the lack of validation, is already acknowledged by the authors.  I am looking forward to seeing the extension of the framework to more advanced tasks.

**Recommended Decision:**

3: Accept

**Relevance:**

3: Solid fit

**Strengths And Weaknesses:**

Strength:

The general research direction of dissentagneling representations based on some prior knowledge of the data is definitely interesting and relevant. The proposed approach is intuitive and reasonable.

Weaknesses:

The approach is validated on a very limited dataset, but given that it is an abstract submission, I don't believe that is a big issue.
The learned embeddings are not visualized, which would help in understanding better the algorithm.


**Submission Track:**

Extended Abstract (4 Page)

---

### Decision · Program_Chairs · 2022-10-21

Accept (Poster)